# Relationship between Changes in Intestinal Microorganisms and Effect of High Temperature on the Growth and Development of *Bombyx mori* Larvae

**DOI:** 10.3390/ijms231810289

**Published:** 2022-09-07

**Authors:** Xiaoning Sun, Qian Yuan, Beibei Du, Xinye Jin, Xiyun Huang, Qiuying Li, Yueqiao Zhong, Zhonghua Pan, Shiqing Xu, Yanghu Sima

**Affiliations:** 1School of Biology and Basic Medical Sciences, Suzhou Medical College, Soochow University, Suzhou 215123, China; 2Institute of Agricultural Biotechnology & Ecology (IABE), Soochow University, Suzhou 215123, China

**Keywords:** silkworm, gut microbes, high temperature, growth and development, material metabolism

## Abstract

Temperature is an important environmental factor affecting the growth and development of silkworm (*Bombyx mori*). To analyze the effect of intestinal microbes on silkworm in response to a high-temperature environment, this study used a combination of high throughput sequencing and biochemical assays to detect silkworm intestinal microbes treated with high temperature for 72 h. The results show that high temperature affects the intestinal microbes of silkworm and that there are sex differences, specifically, females were more sensitive. The changes in the metabolism and transport ability of silkworm intestinal tissues under high temperature are related to the intestinal microbes. High temperatures may affect the intestinal microbes of silkworms, regulating the activity of related digestive enzymes and substance transport in the intestine, thereby affecting the silkworm’s digestion and absorption of nutrients, and ultimately affecting growth and development.

## 1. Introduction

Global warming is likely to pose a serious threat to insect populations through the effects of high temperature on their survival [1,2] and reproduction [3]. High temperature is one of the important reasons for the loss of sericulture production [4,5]. Previous studies have shown that instantaneous high temperature can change the composition of intestinal microorganisms [6]. Microorganisms colonizing an insect’s gut affect their growth, [7] nutrient absorption [8,9], and their resistance to pathogens [10], for example, gut microbial products such as lactic acid can shorten the life span of fruit flies [11]. Temperature is a key factor that affects the growth of silkworm larvae. High temperature affects the food absorption and assimilation of silkworms and shortens the fifth instar growth period [12]. Does high temperature affect the growth and development of silkworms by changing the composition of intestinal microorganisms, thus leading to the loss of sericulture production? Global warming is increasing the frequency of heatwaves and even sustained high temperature events experienced by organisms [13,14], and we focus on the effects on silkworms under extreme high-temperature conditions. Silkworms have a temperature zone that is generally considered to be 7 to 35 °C; the silkworm’s life activities in this temperature zone are normal, and the growth and development speed is appropriate. The sub-lethal high temperature area is generally 35 to 45 °C, which is characterized by poor growth and development. If it lasts for too long, it can cause thermal coma or even death [15]. The last instar of silkworm larvae is the main period of material accumulation, especially from the fifth instar to the fourth day. Therefore, we performed a treatment at 34 °C for 72 h during this period.

The midgut tissue plays a very important role in the digestion, absorption, and transport of nutrients in the silkworm larval stage. The top of the columnar epithelial cells in the midgut tissue forms the brush border membrane to increase the surface area of the intestinal wall cell membrane [16,17], thereby greatly enhancing the organism’s ability to digest food and absorb nutrients. Previous studies have shown that the metabolic process in the intestine is jointly regulated by host genes and microbial genes [18,19]. Microorganisms secrete various carbohydrate-metabolizing enzymes to help the host use a variety of polysaccharides [20,21] and carbohydrates to produce large amounts of fatty acids, which are reabsorbed by the large intestine as a source of energy for the host [22]. Intestinal microorganisms can also decompose proteins. The decomposition products include amino acids that enter the bacterial cells under the action of specific transporters [23]. These products can also directly enter the metabolic pathways through fermentation of a large number of anaerobic bacteria and ultimately produce short-chain fatty acids, organic acids, and ethanol [24]. Previous studies have shown that gut microorganisms, as an “internal environmental factor,” can accumulate fat in the host by regulating the expression of host genes [25,26] and can affect host energy metabolism [27]. However, an impairment of the homeostasis between the intestinal microorganisms and the host’s internal environment can cause the host’s metabolic process to become abnormal [28], thus affecting the growth and development of the host. In the present work, the changes in the composition of intestinal microorganisms of silkworm in a high-temperature environment were studied to determine the relationship between the high temperature resistance ability of silkworm and the gut microorganisms; moreover, the correlation of environment and intestinal microbial-host material metabolism was investigated by determining the effects of a high-temperature environment on intestinal microorganisms of silkworms of different sexes.

## 2. Results

### 2.1. High Temperature Affects the Growth and Development of Silkworm Larvae

We treated the fifth instar larvae for 72 h at 34 °C (Figure 1A). During the fifth instar growth period, the growth and development of silkworms had significant sex differences. Female silkworms were heavier than males in the same developmental period. This gender difference still existed after the high-temperature treatment (Appendix A). During the high-temperature treatment, the food intake (female24 h, *p* < 0.001; 48 h, *p* < 0.001; 72 h, *p* = 0.042. male0 h, *p* = 0.015, 24 h, *p* = 0.007; 48 h, *p* = 0.007) and digestion (female24 h, *p* = 0.031; 48 h, *p* < 0.001. male24 h, *p* = 0.004; 48 h, *p* = 0.004) of the silkworm larvae increased significantly (Figure 1C,D), resulting in the silkworm’s weight during the high-temperature period being significantly higher than that of the control group (female24 h, *p* = 0.027; 48 h, *p* < 0.001; male24 h, *p* = 0.043; 48 h, *p* < 0.001) (Figure 1B). After 72 h of high-temperature treatment, the silkworms were reared in the same environment as the control group (recovery period). Because of the influence of the high temperature in the early stage, the food intake (female96 h, *p* = 0.042; 120 h, *p* = 0.030. male120 h, *p* = 0.024; 144 h, *p* = 0.022) and digestion (female96 h, *p* = 0.009; 120 h, *p* = 0.044. male96 h, *p* = 0.044; 120 h, *p* = 0.013; 144 h, *p* = 0.001) of silkworms during the recovery period were significantly lower than those of the control group (Figure 1C,D). After 72 h of high-temperature treatment, the body weights (female120 h, *p* = 0.048; 144 h, *p* = 0.043. male122 h, *p* = 0.043; 144 h, *p* = 0.002) were significantly lower than that of the control group (Figure 1B). After the high-temperature treatment, silkworms that did not die were raised until cocoon formation, and the percentages of dead silkworm cocoons and waste cocoons were higher than those of the control group, although not statistically significant (Appendix A). This shows that high temperature affects the health and growth of silkworm.

### 2.2. High Temperature Affects Growth and Development by Changing the Morphology and Structure of the Midgut of the Silkworm and the Activity of Related Digestive Enzymes

High temperature could cause changes in the morphology of midgut cells and cause irreversible damage to the midgut of silkworm larvae. The results of HE staining show that the midgut epithelial cells were damaged after 72 h of high-temperature treatment, and the intestinal cells were arranged more loosely and chaotically (Figure 2A). The vacuolation in the midgut section of the CK group and the HT group was quantitatively calculated. In the HT group, both males and females had different degrees of midgut cell vacuolation (Appendix A), and the expression of the silkworm membrane transporter *BmTmC27* gene occurred (Figure 2B,C). This may disorder the material transport homeostasis in the midgut.

We then tested the activity of nutrient-metabolizing enzymes in the intestinal digestive juice of the silkworm. The results show that the activity of trehalase in the intestinal digestive juice of the silkworm was significantly reduced after the high-temperature treatment for 24 h (female, *p* = 0.006; male, *p* = 0.031). Lipase activity (female24 h, *p* = 0.003. male24 h, *p* < 0.001; 48 h, *p* = 0.034; 72 h, *p* = 0.012) was significantly reduced after high-temperature treatment for 24 h, and also significantly reduced in males for 48 h and 72 h (Figure 2D). There was no significant difference in protease activity after high-temperature treatment (Appendix A), and the protein content in the midgut decreased with the increase in high-temperature-treatment time (Figure 2D). High temperature may affect the growth and development of the silkworm by changing the morphology of the midgut and the activity of related digestive enzymes.

### 2.3. High-Temperature-Induced Changes in the Composition and Function of the Intestinal Microbiota of Silkworm Larvae Are One of the Main Causes of Abnormal Digestion, Absorption, Growth, and Development

In order to study the important role played by the gut microbiota in the high-temperature stress of silkworms, we sampled the silkworm larvae treated with high temperature for 72 h at the fifth instar stage, analyzed the microbiota composition by 16S rRNA sequencing, and then performed principal coordinate analysis based on the weighted UniFrac distance. The results show that the silkworm gut microbiota caused by high temperature was quite different from that of the control group (Figure 3A). At the same time, it was found that the number of effective reads obtained by sequencing was reduced, and the effective reads were classified into different operational taxonomic units (OTUs) based on 97% similarity. The results show that high temperature caused a decrease in the number of OTUs of the silkworm intestinal microbiota. The α diversity index (including ACE, Chao1, Shannon, Simpson) also shows that high temperature reduces the richness and diversity of the silkworm gut microbiota (Appendix A).

At the phylum level, Proteobacteria, Firmicutes, Bacteroidetes, and Acidobacteria in the silkworm intestine are the main microbiota (average 50.51%, 28.87%, 9.10%, and 7.82%, respectively) (Figure 3B). High temperature caused a significant decrease in the abundance of Acidobacteria, Fusobacteria, Actinobacteria, and Bacteroidetes in the high-temperature female group compared with the control group. Actinobacteria were significantly reduced in females compared with males after high temperature (Figure 3C).

At the genus level, high temperature could cause a significant decrease in the abundance of 25 species of microorganisms in the intestines of female silkworms (Figure 4A), while 12 species of microorganisms in the intestines of male silkworms significantly decreased (Figure 4B). The 3 types of male–female different microorganisms (Figure 4C) increased to 15 types (Figure 4D) after the high-temperature treatment, and these different microorganisms appeared to have a significant advantage in the male group. The results show that the response of the silkworm gut microbiota to high temperature is manifested as a decrease in microbial diversity and changes in abundance of various microorganisms, and that high temperature increases the difference between males and females.

In order to understand the relationship between the changes in the intestinal microbes of the silkworm and the host function, we carried out “closed-reference OTU picking” on the obtained silkworm intestinal microbial gene sequence, based on the full-length sequence of the tested microbial genome gene. We inferred the gene function spectrum of their common ancestor and that of other unpredicted species in the Green genes database. We constructed the gene function prediction spectrum of the entire lineage of archaea and bacteria. Finally, the bacterial population was obtained by sequencing map to the database to predict microbial function. The prediction results show that the function of silkworm gut microbes is mainly related to metabolism, organismal system, environmental information processing, cellular processes, human diseases, and genetic information processing. High temperature causes the proportion of silkworm gut microbes to participate in environmental information processing and metabolism lower than that of the control group (both female and male) (Figure 5A). In the KEGG secondary metabolic pathway, the high-temperature treatment group showed that the functions of signaling molecules and interaction, transport and catabolism, lipid metabolism, and biosynthesis of other secondary metabolites, as well as neurodegenerative disease and cardiovascular diseases, were significantly downregulated compared with the control group (Figure 5B). In addition to these, high temperature also caused the female silkworm gut microbes to participate in the membrane transport function. The male silkworm participated in the significant upregulation of translation, replication and repair, digestive system, and metabolic diseases. The male silkworm participated in the metabolism of other amino acids, cell growth, and downregulation of six functions including death and circulatory system (Figure 5B). It can be seen that high temperature causes changes in the function of silkworm intestinal microbes and that there are sex differences in such changes. The analysis shows that there are significant sex differences in the intestinal microbial functions of the silkworm, which are mainly manifested in metabolism, organismal system, human diseases, and genetic information processing.

Correlation analysis between digestive enzyme activity and differential microbial abundance was performed. It can be seen from Figure 6A that there were eight kinds of microorganisms that are significantly related to THL activity in the intestines of female silkworms in the control group (seven kinds are positively correlated, and one kind is negatively correlated). There are also five kinds of microorganisms that are significantly correlated with THL activity in the intestines of male silkworms (all are positively correlated), and these microorganisms that are significantly related to THL activity are different in the male and female groups. After high-temperature treatment, the number of microorganisms significantly related to THL activity in the female silkworm’s intestine rose to 16 (all positively correlated), while there were no microorganisms significantly related to THL activity in the male silkworm’s intestine. These microorganisms are different between the male and female groups and are different from the control group.

It can be seen from Figure 6B that there are two kinds of microorganisms that are significantly related to LPS activity in the intestines of female silkworms in the control group (one is positively correlated, and the other is negatively correlated), and that there are seven kinds of microorganisms that are significantly correlated with LPS activity in the intestines of male silkworms (four species are positively correlated, and three species are negatively correlated). These microorganisms that are significantly related to LPS activity are different in the male and female groups. After high-temperature treatment, the number of microorganisms significantly related to LPS activity in the female silkworm’s intestine was reduced to one (Brachybacterium was positively correlated with LPS activity), and the number of microorganisms significantly related to LPS activity in the male silkworm’s intestine was reduced to one (Paenibacillus and LPS activity were negatively correlated). Similarly, these microorganisms were different between the male and female groups and were different from the control group.

In order to explore whether the effect of high temperature on the digestion and absorption of silkworms is related to intestinal microbes, we further analyzed the flora related to material and nutrient metabolism. The results show that high temperature caused a significant decrease in the abundance of various microbes in the intestinal tract of female silkworms. Among them, Clostridium involved in carbohydrate metabolism was significantly reduced (Figure 6C); additionally, Propionibacterium and Fusobacterium, which are related to protein and amino acid catabolism in the small intestine, were significantly reduced. The males only showed differences in Fusobacterium (Figure 6A). It is speculated that high temperature may further affect digestion and absorption by affecting the flora related to material and nutrient metabolism, and that it shows certain sex differences.

## 3. Discussion

### 3.1. High Temperature Affects Silkworm Growth and Development by Changing the Intestinal Microbial Flora

Gut microbes are a complex food web that can digest and utilize nutrients in the diet [29,30,31]. Bacteroides and other microorganisms can use carbohydrates in the intestine to produce large amounts of fatty acids, which are reabsorbed by the large intestine as a source of energy for the host [22]. Clostridium cluster IV and XIVa can ferment dietary carbohydrates to produce butyric acid, Clostridium cluster IX can convert lactate to propionate [32], and some species of Clostridium cluster XIVa can use carbohydrates to convert to butyrate [33]. These large amounts of short-chain fatty acids produced by microbial fermentation, including acetic acid, butyric acid, formate, propionate, and acetate, have important physiological functions for the host, such as improving the intestinal environment and providing energy to the host [34,35]. In order to further clarify the reasons for the growth and development of silkworms under high-temperature conditions, this paper studied the changes in intestinal flora before and after high-temperature treatment of males and females based on high-throughput sequencing of 16S rRNA genes. After the high-temperature treatment, the abundance of Clostridium significantly decreased, affecting the host’s utilization of carbohydrates, and the decomposed short-chain fatty acids decreased accordingly. This may be one of the reasons for the significant decrease in the body weight of the silkworm after the high-temperature treatment compared with the control group. The mechanism needs to be further studied.

### 3.2. The High-Temperature Treatment Did Not Cause a Significant Change in Protease Activity, but the Protein Content of Lipase Was Significantly Reduced

We speculate that this may be caused by the damage to the silkworm intestinal tissue in the high-temperature environment, which affects the permeability of the intestinal wall [36]. Through the analysis of the above results, we believe that high temperature regulates the activity of related digestive enzymes in the intestines and tissue transport capacity by affecting the changes in the intestinal microbes of the silkworm, thereby affecting the digestion and absorption of nutrients by the silkworm. Studies have shown that high temperature can cause intestinal damage and dysfunction [37]. The investigation found that Gemmatimonadetes and Firmicutes were significantly negatively correlated with the transcription level of the membrane transporter gene *BmTmC27*. It is speculated that high temperature may affect the material transport ability of the midgut tissue. However, which microorganisms are involved in the membrane transport function of the silkworm and the specific impact mechanism still need to be further studied.

### 3.3. There Are Sex Differences in the Effects of High Temperature on the Intestinal Microbes of the Silkworm

As the first line of defense of silkworm innate immunity, midgut epithelial cells are the initiating organs of systemic inflammatory response syndrome and even multiple organ dysfunction syndrome [38,39]. Studies have shown that long-term exposure of silkworms to high-temperature conditions can cause viral disease in silkworms [5,40], affecting production performance, with females being less resistant to high temperatures than males [41]. Growing research shows sex differences in insect immunity, such as sex-specific mortality of *Lymantria dispar* L. after larval infestation by the entomopathogenic bacterium *Bacillus thuringiensis* berlin. Interestingly, only the antibacterial activity of the female midgut is activated by infection [42]. Phenol oxidase (PO) is often used to estimate insect resistance to immune insults. Significant sex differences in PO activity were found in the yellow dung fly *Scathophaga stercoraria* [43]. In *Nilaparvata lugens*, female adult worms possess much more microbial symbionts than male adults. This study suggests a possible immune strategy whereby female adult worms maintain microbial symbionts by reducing autoimmunity to meet nutritional, developmental, and reproductive needs [44]. From the results of this study, we found that before and after the high-temperature treatment, the abundance of silkworm intestinal microbes showed sex differences at both the phylum and genus levels. This sex difference may be one of the reasons for the difference in resistance to high temperature between males and females, which will be studied further.

## 4. Materials and Methods

### 4.1. Experimental Animal

The classic genetic and developmental research silkworm strain, *P50*(*Dazao*) [45], was employed in this study and maintained in the School of Biology and Basic Medical Sciences, Medical College of Soochow University. The larvae of silkworm were reared on fresh mulberry leaves under 12 h light (66 lux)/12 h dark conditions at 25 ± 1 °C with 65 ± 5% humidity during the non-high-temperature treatment period. We picked larvae of similar weight, and separated males and females at the beginning of the fifth instar. The fifth-instar larvae fed for 12 h and were fed at a high temperature (34 ± 1 °C) for 72 h to extract intestinal microbes. Larvae reared at 25 ± 1 °C were used as controls.

### 4.2. DNA Extraction and High-Throughput Sequencing

In a sterile operating table, the epidermis of the freeze-killed silkworm was sterilized with alcohol, then placed in PBS to dissect the complete intestine, and placed in a pre-cooled 5 mL sterile centrifuge tube, quick-frozen in liquid nitrogen, and stored at −80 °C. The intestinal contents of every 10 silkworms were mixed as one intestinal microbe extraction sample, and 5 samples were repeated for each treatment group. We used the Omega Stool DNA Kit (OMEGA, Atlanta, GA, USA) to extract the genomic DNA of intestinal bacteria by referring to the instructions. We stored the extracted DNA in a refrigerator at −80 °C. Shanghai Personal Biotechnology Co., Ltd. (Shanghai, China) conducted follow-up sequencing (Illumina MiSeq, San Diego, CA, USA) and bioinformatics analysis. The process is to use Illumina’s TruSeq Nano DNA LT Library Prep Kit for library construction, using primers from the V3–V4 region of bacterial 16S for amplification: 338F primer, 5′-ACTCCTACGGGAGGCAGCA-3′, 806R primer, 5′-GGACTACHVGGGTWTCTAAT-3′.

### 4.3. Data Analysis

Sequencing raw data were saved in the Fastq format, and the sliding window method was used for quality screening. FLASH software was used to pair and connect according to the overlapping bases [46]. The effective sequence of each sample was obtained according to the barcode sequence corresponding to each sample. Questioning sequences were identified using QIIME software [47]. The length of the sequence was required to be ≥150 bp, and the ambiguous base N was not allowed. Sequences with >1 mismatched bases at the 5′-end of primers and sequences containing consecutive identical bases were eliminated. Sequences with >8 bases were checked by Usearch to eliminate chimera sequences and obtain optimized sequences.

The Uclust sequence alignment tool [48] was used on the obtained optimized data, and the operational taxonomic unit (OTU) was calculated according to the sequence similarity of 97%; the sequence with the highest abundance in each OTU was selected as the representative sequence of the OTU. By using the QIIME software, the total number of sequences in each sample in the OTU abundance matrix was randomly sampled at different depths, and by randomly sampling a certain number of sequences from each sample, the samples were predicted to be contained in a range of given sequencing depths [49]. A rarefaction curve was drawn for the total number of species and the relative abundance of each species (Appendix A). Alpha diversity index calculation results are shown in Appendix A.

The Simpson index, Chao index, Shannon index, and Good’s coverage were calculated using Mothur to estimate the genera richness and α diversity of each sample. Significant taxonomic differences between the samples were also calculated by Mothur. The Blastn algorithm was used to compare species richness between different treatments, and QIIME was used to randomly choose a series of subsets of each library in different sizes to calculate respective richness indices.

### 4.4. Quantitative Real-Time PCR

Total RNA was isolated from the midgut tissue at 24, 48, and 72 h after the high-temperature treatment by using RNAiso Plus (TaKaRa, Dalian, China). The cDNA was synthesized using a PrimeScriptTM RT Reagen Kit with gDNA Eraser (TaKaRa, Dalian, China), according to the manufacturer’s instructions. Quantitative real-time reverse transcription PCR (RT-PCR) was used to analyze the mRNA transcript levels of *BmTmC27* gene. The Bm18S rRNA gene was used as an internal control. RT-PCR was performed using a total reaction volume of 20 µL with an ABI StepOnePlusTM Real-Time PCR System (Applied Biosystems, San Francisco, CA, USA) and the fluorescent dye SYBR Premix Ex Taq (TaKaRa, Dalian, China), according to the manufacturer’s instruction and the method described by Ji et al. Bm18sRNA, forward primer 5′ CGATCCGCCGACGTTACTACA 3′ and reverse primer 5′ GTCCGGGCCTGGTGAGATTT 3′. *BmTmC27* gene forward primer 5′ CTTGTGTAGCACTAGCTTCAATTTC 3′ and reverse primer 5′ AAGTGAGCCCCATAAAGCAAAATAA 3′.

### 4.5. Hematoxylin-Eosin Staining (HE Staining)

After the fifth-instar larvae of silkworm were reared at room temperature for 12 h, they were reared at room temperature and high temperature for 72 h. The midgut tissue was dissected and washed with normal saline. The tissue was then completely immersed in 4% paraformaldehyde solution to fix it, and then it was successively dehydrated in ethanol in the following order of increasing concentration: 30%, 50%, 70%, 85%, 95%, and 100%. Each lasted 15 min. It was then treated with xylene twice, 15 min each time, to make it transparent, and then the treated tissue block was immersed in the paraffin solution, placed in a suitable position with tweezers, and the paraffin was cooled. We used a paraffin microtome to cut the paraffin sections (5–10 μm thickness), placed the tissue sections on a glass slide pre-coated with glycerol, and finally put the slide glass in an oven at 37 °C for 12 h.

### 4.6. Determination of Activity of Intestinal Digestive Juice Enzyme

The silkworms treated at high temperature for 24, 48, and 72 h were used to extract digestive solution. The silkworms were placed on a foam board and fixed with a dissecting needle. The body wall was longitudinally cut to expose the digestive tract, and other tissues attached to the digestive tract were removed. Hemolymph was washed away with cold PBS. We used sterile filter paper to absorb the residual liquid, took out the intact intestine with tweezers and placed it over the mouth of the centrifuge tube with the front end facing down, then cut the front-end wall of the intestine so that the digestive solution naturally flowed into the pre-cooled centrifuge tube, and multiple silkworm digestion solutions were combined into a mixed sample. Each sample was centrifuged at 4000 r/min for 10 min at 4 °C to take the supernatant, and guaranteed to have a volume of more than 800 μL. We repeated three samples at each time point. We recorded the number and saved it at −80 °C. After the three recording points were collected, they were processed in a unified manner. The enzyme activity was determined using a neutral protease, trehalase, and lipase kit (Comin, Suzhou, China) (measurement wavelengths: neutral protease and lipase at 710 nm, trehalase at 550 nm). The definition of neutral protease activity unit is as follows: 1 μmol of tyrosine produced by catalytic hydrolysis per ml of sample per minute at 30 °C is 1 enzyme activity unit. Neutral protease activity (μmol/min/mL) = C standard × (A assay tube − A control tube) ÷ (A standard tube − A blank tube) × dilution factor ÷ V1 ÷ T, where C standard: 0.25 μmol/mL standard tyrosine solution; V1: the volume of crude enzyme solution added to the reaction system; and T: catalytic reaction time (min). The definition of lipases activity unit is as follows: one unit of enzyme activity is the hydrolysis of olive oil per milliliter of serum at 37 °C to generate 1 μmol of fatty acids per minute. Lipases activity (μmol/min/mL) = [C standard product × (A measuring tube − A blank tube) ÷ (A standard tube − A blank tube)] × V1 ÷ V2 ÷ T, where C standard product: 10 µmol/mL; V1: total volume of the reaction; V2: the volume of sample added to the reaction; and T: time for catalytic reaction. The definition of trehalase activity unit is as follows: one unit of enzyme activity is defined as the catalytic production of 1 μmol of glucose per mL of sample per minute. Trehalase activity (μmol/min/mL) = [1000 × (ΔA + 0.6264) ÷ 0.6555 × V1] ÷ (V3 × V1 ÷ V2) ÷ T ÷ 180, where V1: volume of sample added; V2: volume of extract added; V3: volume of sample added; and ΔA = A assay tube − A control tube (one control tube is required for each assay tube). For the protein content we used the Compatibility Chart for the BCA Kit (Beyotime, Shanghai, China), and measurement wavelengths at 562 nm. Protein content (mg/mL) = C standard × (A assay tube − A blank tube) ÷ (A standard tube − A blank tube).

### 4.7. Protein Content Determination

After the high-temperature treatment for 24, 48, and 72 h, the silkworm digestion solution (multiple silkworm digestion solutions mixed) was extracted, and three samples were repeated at each time point, and the number was recorded and stored at −80 °C. After the three recording points were collected, the enzyme activity was determined using the protein content determination kit (Comin, Suzhou, China) (measurement wavelength: 562 nm).

### 4.8. Growth and Development Index Survey

The fifth-instar silkworms were reared in male and female divisions, with 30 larvae in each division, and three groups were repeated. After 12 h of feeding, control groups were raised in a constant-temperature incubator at 25 °C, and high temperature groups were raised in another constant-temperature incubator at 34 °C. Each group was fed 3 times a day at 8:00 am, 2:00 pm, and 8:00 pm. The amount of mulberry was recorded before each feeding, and the weight of the silkworms, the number of dead silkworms, the dry weight of mulberry leftover, and the dry weight of silkworm manure were recorded every 24; additionally, the cocoon quality was investigated after cocooning. The calculation formula is as follows: Food intake = (giving mulberry dry weight − leftover mulberry dry weight)/number of larvae per group; digestion = (ingestion amount − dry weight of silkworm manure)/number of larvae per group. The larvae of each treatment group were fed to cocooning, and the number of cocoons was counted (Total cocoon). On day 6 after the cocooning, the number of deaths in the cocoons (Dead worm cocoon), number of spoiled cocoons (Spoiled cocoon), and number of common cocoons (Common cocoon) were investigated, respectively. The proportion of dead-worm cocoons (%) = Dead worm cocoon/Total cocoon. The proportion of spoiled cocoon (%) = Spoiled cocoon/Total cocoon. The proportion of common cocoon (%) = Common cocoon/Total cocoon. Student’s *t*-test and one-way ANOVA were used to analyze statistical significance.

### 4.9. Statistical Analysis

GraphPad Prism version 8 (GraphPad, San Diego, CA, USA) was used for statistical computations and graph construction. Data are presented as the mean ± SD. Significance analysis was performed using *t* test and corrected by Holm-Sidak method, *p* < 0.05 was accepted as significant.

## 5. Conclusions

High temperature affected the intestinal microbial composition of silkworms; specifically, the intestinal microbial abundance of female silkworms was significantly reduced. The changes in intestinal tissue damage, membrane transport function, and material metabolism functions of silkworms under a high temperature environment are related to the changes in intestinal microbial composition and function.

## Figures and Tables

**Figure 1 ijms-23-10289-f001:**
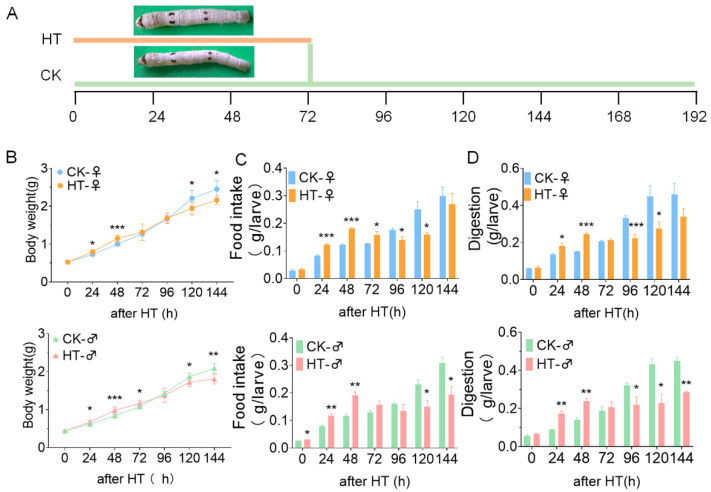
The effect of high temperature on the growth and development of silkworm. (**A**) Time axis of high-temperature treatment. (**B**) The weight survey of silkworm, *n* ≥ 7. (**C**) Food intake capacity of the larvae (grams per larva). (**D**) Digestion capacity of the larvae (grams per larva). *, *p* ≤ 0.05; **, *p* ≤ 0.01; ***, *p* ≤ 0.001. CK, control group; after HT, high-temperature treatment.

**Figure 2 ijms-23-10289-f002:**
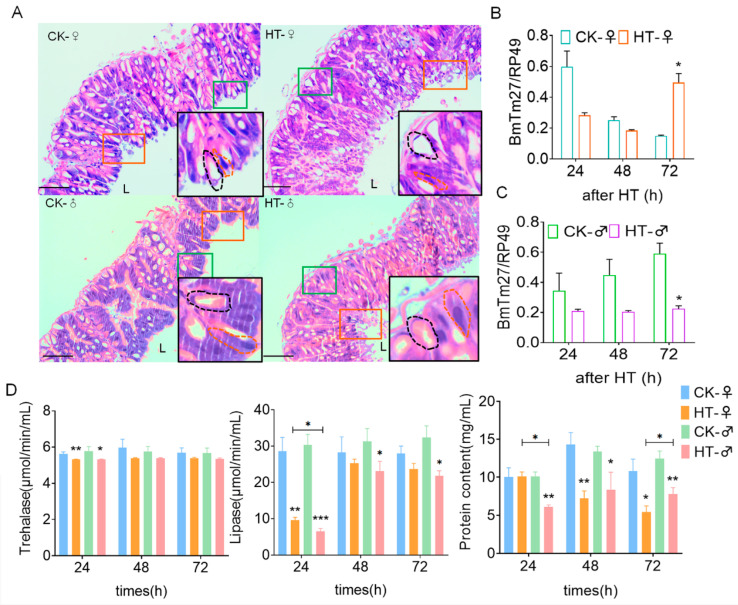
The effect of high temperature on the digestion and absorption in silkworm. (**A**) HE staining of a section of the silkworm midgut tissue. The midgut tissue was taken out after the silkworm was treated with high temperature (34 °C) for 72 h. The red dotted line represents cylindrical cells, the black dotted line represents goblet cells, and L represents the intestinal lumen. Bar = 100 μm. (**B**) Transcription level of the *BmTmC27* gene in midgut tissue of female silkworm. (**C**) Transcription level of the *BmTmC27* gene in midgut tissue of male silkworm. (**D**) The activity and protein content of lipase and trehalase in digestive juice during high-temperature treatment. *, *p* ≤ 0.05; **, *p* ≤ 0.01; ***, *p* ≤ 0.001. CK, control group; after HT, high-temperature treatment.

**Figure 3 ijms-23-10289-f003:**
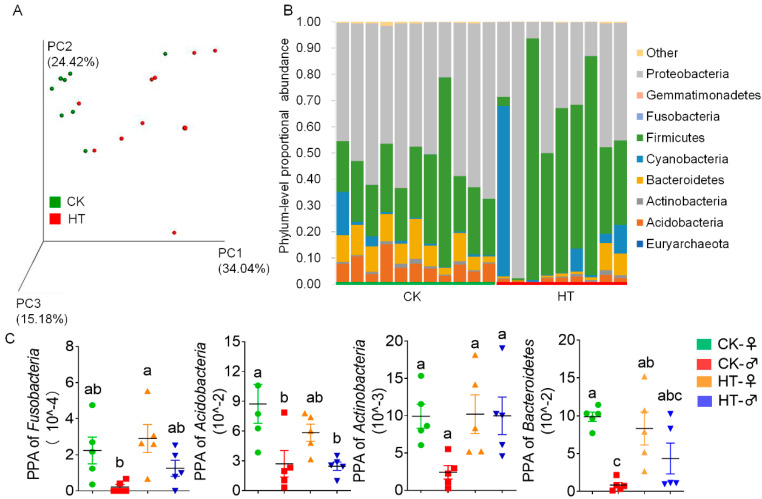
Composition and abundance of gut microbes of silkworm after the high-temperature treatment. (**A**) Analysis of sample principal coordinates. The percentage in the brackets of the coordinate axis represents the proportion of the difference in the original data that can be explained by the corresponding principal coordinate. Principal components (PCs) 1, 2, and 3 are the characteristic values that cause the differences in samples in each group, which are 34.04%, 24.42%, and 15.18%, respectively. (**B**) The gut microbial composition of the sample at the phylum level. (**C**) The abundance of gut microbes of the sample at the phylum level. Bars with different letters are significantly different (*p* < 0.05).

**Figure 4 ijms-23-10289-f004:**
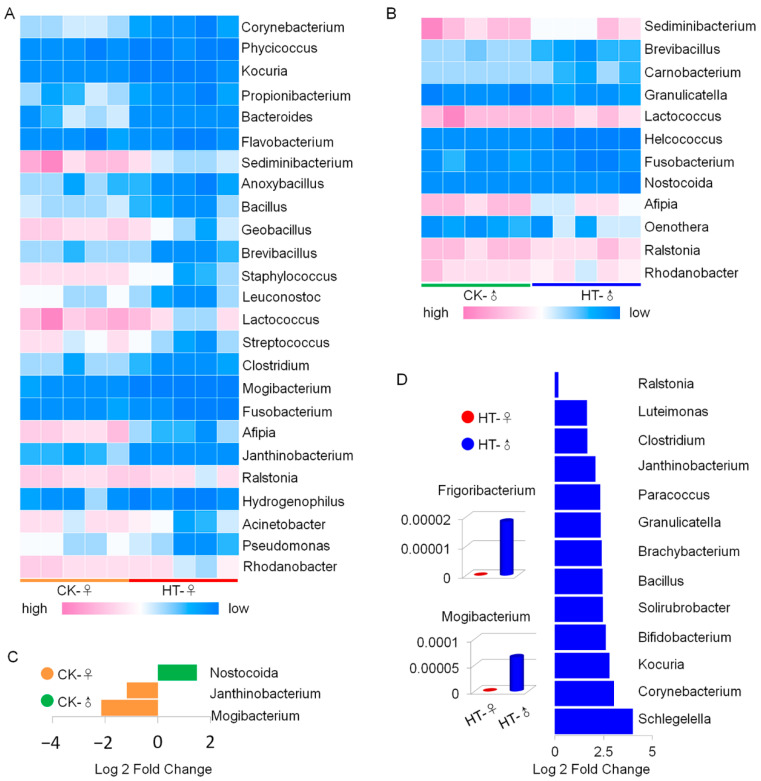
Differential microbes in the microbial genera levels in the silkworm gut caused by high temperature. (**A**) Microbes at the genus level. High temperature caused significant differences in female silkworms (*p* < 0.05). (**B**) At the genus level, high temperature caused significant differences in male silkworms (*p* < 0.05). (**C**) At the genus level, there were significant differences in the intestinal microbes of the silkworm in the control group (*p* < 0.05). (**D**) At the genus level, there were significant differences in the microbes in the silkworm intestines in the high-temperature group (*p* < 0.05). The bar plot shows gut microbiota at the genus level that were significantly enriched (*p* < 0.05) as determined by one-way ANOVA. Fold change of the main discriminant between CK-♀ and CK-♂; fold change of the main discriminant between HT-♀ and HT-♂.

**Figure 5 ijms-23-10289-f005:**
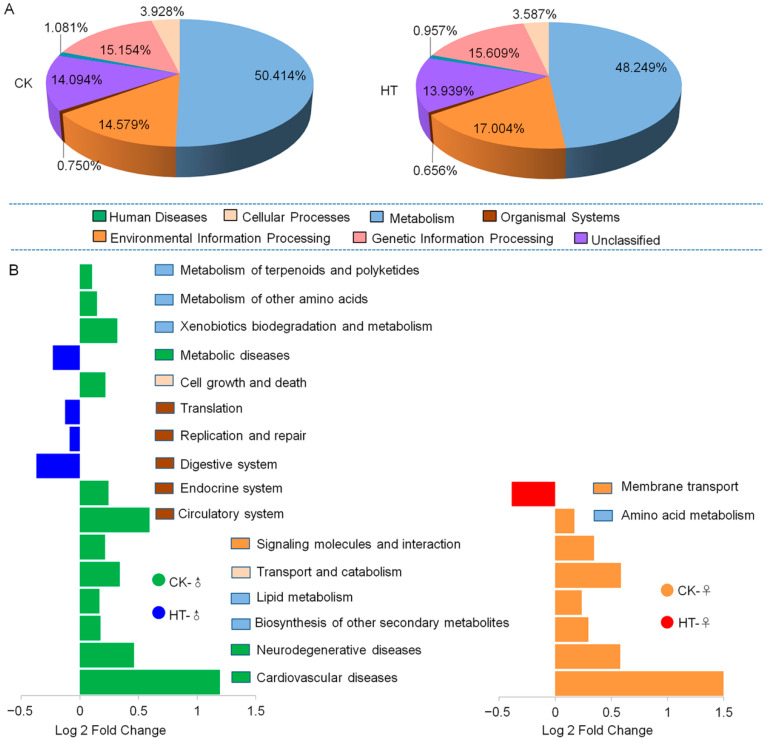
KEGG analysis of gut microbes of silkworm. (**A**) DistributiEW2on of primary metabolic pathways of intestinal microbes KEGG (Kyoto Encyclopedia of Genes and Genomes). (**B**) There are significant differences in the intestinal microbial KEGG secondary metabolic pathways between female and male silkworms in high-temperature environments. The bar plot shows pathways that were significantly enriched (*p* < 0.05) as determined by one-way ANOVA. Fold change of the main discriminant between CK-♀ and HT-♀; fold change of the main discriminant between CK-♂ and HT-♂.

**Figure 6 ijms-23-10289-f006:**
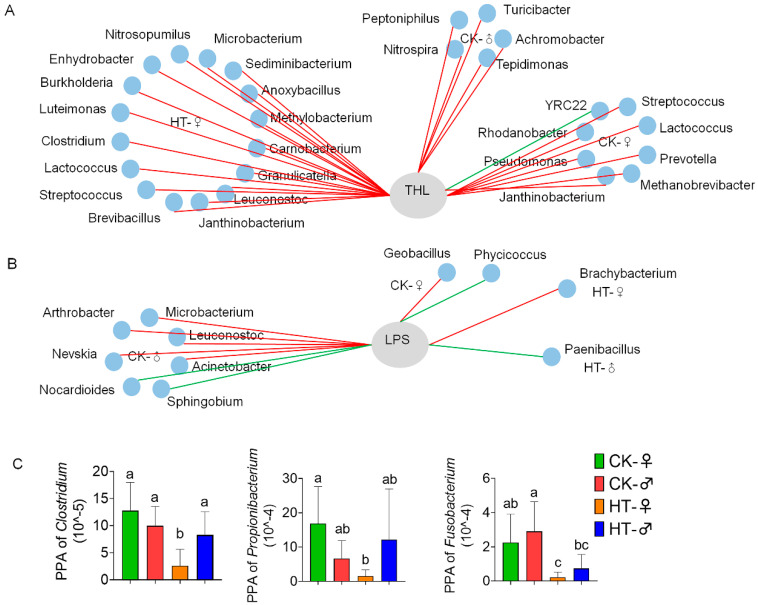
High temperature affects digestive enzymes and nutrient metabolism through the silkworm gut microbes. (**A**) Analysis of the correlation between THL activity in the silkworm digestive juice and the intestinal microbes of the silkworm. (**B**) Correlation analysis between LPS activity in silkworm digestive juice and intestinal microbes. Each co-occurring pair among the microbial populations and metabolites has an absolute Spearman rank correlation above 0.90 (red line: positive correlation (R > 0.90); green line: negative correlation (R < −0.9)) with an FDR-corrected significance level under 0.5. (**C**) Changes in the abundance of bacteria at the genus level after 72 h high-temperature treatment. PPA stands for phylum-level proportional abundance. Bars with different letters are significantly different (*p* < 0.05).

## Data Availability

Not applicable.

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
