# Peer review of "Relationship between Changes in Intestinal Microorganisms and Effect of High Temperature on the Growth and Development of *Bombyx mori* Larvae"

_ijms, 2022, doi:10.3390/ijms231810289_

Round 1
Reviewer 1 Report
The results obtained in this work don't really deserve to be published in this high level journal.
Author Response
Thank you very much for your comments concerning our manuscript.
Reviewer 2 Report
The considered MS is directed to the study the mechanisms of effect of high temperature on economically important insect species. This study is solid, match with the aims and scopes of IJMS and use appropriate methods to reach main task. I would especially estimate that authors study sex dependence in physiological reactions that significantly increase the merit of this study. Generally I think this MS have serious potential for publication. One major comment relate with insufficient justification of experimental design, that generate following technical questions. In particular, authors justify the studding of high temperature by global climate change. But this is too general phenomenon. Which aspect of this phenomenon author would like to test: annual increasing of degree-days that is crucially important for invertebrates or climate change triggered heat waves (that is most close to the experimental design of this study) or something else? This information will help to reader to understand why authors used so high temperature, why exposition was assign for only 3 days, etc. In other words, what situation mimic this experiment? The ideal section for this purpose is first paragraph of methods, where author could justify the experimental design, describe it in detail (ideally referring to scheme where all experiments will be shown, including sample size for each parameter). Last moment is important because it will help to reader understand how you used some parameter to calculate correlation (i.e. parameters must be measured from same individuals). Even this is physiological study, the skeleton is ecological and experimental design is very important here.
Other minor/technical comments which could make this MS more informative you could find below:
11) Abstract. I think that the result that female is more sensitive sex to the temperature in the context of gut microflora diversity is very important and should be included in the abstract.
22) In the section 4.1 authors should more clear indicate what temperature was used for rearing of larvae until high temperature treatment. I guess that 25 C (according to control) but need more clear statement in MS.
33) In the section 4.2 author should more clear describe the procedure of gut dissection (did they make it in PBS or without? Did they divide gut content from midgut tissue? Were samples immediately frozen? Were larvae surface sterilized before dissection?). Also there is no information about library preparation and sequencing platform even it was done in commercial service. I think authors need brifly mention this information because some companies have several kinds of facility for sequencing.
44) Section 4.3 did authors clear whole reads from chloroplasts which usually significantly contaminate the samples of leaf feeding animals?
55) 351-352 from midgut tissue?
66) Section 4.6 Could you describe in more detail how you extract digestion solution? Being the entomological physiologists it is not trivial task even for me. Auditory of IJMS is much wider then entomologists, so much better provide exact protocol or reference for the extraction. Could you also proved the references for enzyme measuring to make it easily replicable by other researchers. What unit was used (do you used enzyme standard?) and how it was normalized (per value or per mg of protein)?
77) Section 4.9 Duncan test? I think this paragraph has too concise description. In the result section we could find PC analysis, correlations, etc… so it have to be described in the methods. also authors should describe the type of distributions of samplings before using appropriate method of analysis.
88) L 67 and in other places. Why authors put the accent for LONG period of high temperature effect? For example to me, long effect is when most part of larvae stage developed under some treatment. In fact, effect was occurred only on the part of fifth instar. So please describe in detail why you relate 3 day treatment to long effect or remove this accent from the text. This question relate to the major comment described above, about the context of temperature effect on B mori. If author will explain in more detail what situation they would mimic by three days +34 effect, maybe this comment will be canceled.
99) Results. I am not too strict regarding statistics results presenting, but often (especially in ecological studies) reviewers want to author provide full statistics metrics (F, df, etc) not only p value. To me, if authors will carefully respond on comment #7 it would be enough but it could be asked by another reviewers.
110) L 83-85 Again, no description of which statistical method was used for survival analysis. As well as no statistical results, confirming the significance of increased mortality/waste cocoon.
111) Figure 2 (Enzyme activities). This is formal comment but anyway. Y-axis mean ACTIVITIES of listed enzymes. Regarding abbreviation – I would suggest to use full enzyme title because it is informative and do not lead to confusing such as LPS (lipases or lipopolysaccharide), or Cpr (no decoding in the text). Also, did authors check the samplings for outliers/extremes? Treated males at 24 hr? I guess there is variable in the sampling does not belong to presented sampling.
112) L 102 destruction is not suit word together with homeostasis. Maybe unbalancing of … ?
113) Does in figure 5a results combine both sexes? Please clarify
114) Section 3.3. I do surprised that author give an examples of remote animal species about sex-specificity in physiology. Please see Belousova et al. 2021, Journal of insect science and references within there. The title of the study is “Sex Specifcity in Innate Immunity of Insect Larvae”. There are many studies (at least not few) where researchers study sex dependence and insect physiology.
115) I guess some kind of conclusion (main results) should be in the end of this solid study.
Generally, described above demonstrate mostly technical issues, not issues with matter/ inappropriate methods. Thus the decision is major revision which should not take a lot of time for revising by authors.
Reviewer 3 Report
.
Round 2
Reviewer 2 Report
Authors considerably revise the MS and significantly improve it. Only one thing is still unexplained. In comment # 10 I asked to mention which statistical method was used for comparing the survival. After this technical information the ms could be accepted.
Author Response
请参阅附件。
